# Can Vitamin D Levels Alter the Effectiveness of Short-Term Facelift Interventions?

**DOI:** 10.3390/healthcare11101490

**Published:** 2023-05-20

**Authors:** Daniela Florina Trifan, Adrian Gheorghe Tirla, Andrada Florina Moldovan, Calin Moș, Florian Bodog, Teodor Traian Maghiar, Felicia Manole, Timea Claudia Ghitea

**Affiliations:** 1Faculty of Medicine and Pharmacy, Doctoral School, University of Oradea, 410068 Oradea, Romania; 2Faculty of Medicine and Pharmacy, Medicine Department, University of Oradea, 410068 Oradea, Romania; 3Faculty of Medicine and Pharmacy, Department of Pharmacy, University of Oradea, 10, 410068 Oradea, Romania

**Keywords:** minimally invasive surgery, vitamin D, facelift, combined techniques

## Abstract

Facelifting is increasingly popular among the population. It exceeded the limits of post-traumatic facia-reconstruction. Both the demand and the methods available are getting increasingly diverse. The minimally invasive technique revolutionized the facelift, although it took some time to completely comprehend the mechanics. The roles of vitamin D in numerous physiological processes in which it is involved have mostly been elucidated in the last decade. Our hypothesis is based on one of these roles, that is, vitamin D intervenes in changing the type of collagen by changing its location; therefore, collagen will have a supporting role for the subcutaneous tissue. A group of 156 patients with different facelifting methods was followed: 93 minimally invasive (NC), 49 classical surgery (C) and 14 with the combined technique (NC + C). The change in the subcutaneous tissue was monitored by an elastograph. The level of vitamin D was monitored in order to assess the immediate and long-term effects of vitamin D on the progression of subcutaneous fibrosis. It was proven that an optimal level of vitamin D has a beneficial effect in maintaining the volume of subcutaneous tissue in patients from the NC and NC + C groups, the best results being in the NC + C group. An increase in the subcutaneous volume was recorded, which leads to a decrease in elasticity (statistical significance *p* < 0.05) and the lowering of the subcutaneous tissue, and an increased amount of lowering corresponds to a lowering of vitamin D levels.

## 1. Introduction

Aging skin exhibits structural, cellular and molecular changes and an accumulation of senescent cells [1]. Perioral aging is characterized by fine lines, marionette lines, the flattening of the cupid’s bow, a long and poorly defined philtrum and a narrow vertical and wider transverse smile. These changes are the result of genetics, photoaging, smoking, gravity, repetitive pursing of the orbicularis oris, changes in dentition and bone resorption [2,3] at the maxillo-mandibular level. Jaw resorption can also contribute to the loss of upper lip support, resulting in the formation of perioral wrinkles, commonly known as “smoker lines” [4]. The facelift ideally repositions the ptosis, superior soft tissue, to restore facial volume and a younger appearance. Over time, the facelift has progressed from a simple incision of the skin folds to minimal-access cranial suspension and more sophisticated interventions at the level of the superficial musculoaponeurotic system (SMAS) [5]. The major purpose of facelifting is to assess the efficacy and safety of lifting interventions among patients with varying degrees of ptosis on the face [6]. Many physiological changes can be slowed with less aggressive treatment methods, although the effectiveness always depends on external risk factors.

The use of autologous fat transfer has greatly reduced the risk of hypersensitivity or allergic reactions, as well as the risk of infectious disease transmission [7]. However, during surgical interventions, especially classical ones, patients are at a particular risk of perioperative complications caused by the anesthetic used during the procedure [8]. The broad spectrum use of minimally invasive treatment has reduced the number of complications significantly. Long-term adverse effects such as persistent granulomas and infections have been reported. These may be due to the inappropriate handling of the materials used intraoperatively or sterile instruments, the body’s reaction to the materials used in the facelift [9].

The complete removal of all remaining fat deposits below the mandibular region and the gonial angle improve the definition of the jawline [10,11]. A 3.5 cm long submental incision is centered over the midline, approximately 5 mm caudal to the submental fold [12]. The dissection is carried out through the subcutaneous tissue, identifying the medial edges of the platysma muscle [13]. It is essential that for the majority of patients who require a facelift, the results are consistently excellent, according to patient expectations [14].

Fibrosis (progressive scarring) is often defined as an exaggerated wound-healing response due to sustained chronic injury, micro trauma and other factors. It is attributed to excessive deposition of extracellular matrix components, mainly collagen [15]. While chronic inflammation typically precedes fibrosis, there are distinct, independent inflammatory mechanisms that regulate fibrogenesis, such as dysregulation of myofibroblastic progenitor cell differentiation, myofibroblastic collagen synthesis and extracellular matrix repair [15,16]. Because vitamin D is a fat-soluble prohormone obtained from food sources or from de novo synthesis in the skin, as a result of UV light-induced photolytic conversion of 7-dehydrocholesterol to previtamin D3 followed by thermal isomerization to vitamin D3 [17], it has an important role in the increase in the antifibrotic factor, with a role in the deposit of collagen in the cell wall (type IV). The level of vitamin D is correlated with the synthesis of several types of collagen [18]. Hypovitaminosis D is a major public health problem. Serum 25-hydroxyvitamin D (25-OHD) is now acknowledged as an independent predictor for cardiovascular diseases, but it also has [18] gastrointestinal [19] and autoimmune diseases [20] reducing actions. Serum levels of vitamin D are tightly regulated on two levels. First, an excess of sunlight (ultraviolet B radiation) leads to photodegradation of vitamin D and the formation of suprasterols [21]. Second, elevated levels of 1.25 (OH) 2D3 stimulate the expression of CYP24A1, which is the 24-hydroxylase and is responsible for the catabolism (deactivation) of both 25 (OH)D3 and 1.25 (OH) 2.D 3 [21,22]. Vitamin D receptor deficiency in the epidermis marks an increased dermal thickness, inflammatory cell infiltration and severe collagen deposition. The increase in the number of vitamin D receptors leads to the overproduction of collagen (Col1A1, Col1A2, Col3A1, α-SMA, MMP9, TGF-β1) and pro-inflammatory cytokines (IL–1β, IL–6, CXCL1, CXCL2) [23].

Depending on the type of intervention (surgical, minim-invasive or mixed), these signs of aging can be corrected in different percentages.

It has been observed that in some cases (a low percentage of 3–10%) patients do not respond to facelifting techniques, regardless of the technique used. Analyzing all clinical and paraclinical parameters, a hypothesis of modified collagen formation was put forward. The objective of this work is to verify this hypothesis.

In this sense, a direct link was found between vitamin D levels and the formation of healing collagen [24,25,26,27,28,29,30,31,32,33,34,35,36]. All patients were given a rapid vitamin D test, and a difference in vitamin D levels was noted. To consolidate the results, additional studies are needed, such as blood vitamin D levels and osteoporosis or chronic inflammatory diseases. If the hypothesis turns out to be true, this would be a new perspective in terms of increasing the effectiveness of the facelift. The present study correlated the reconstruction of different areas with the results obtained, in order to be able to recommend the most efficient lifting technique, personalized for each patient, and the connection between the formation of subdermal fibrosis and the level of vitamin D is followed.

## 2. Materials and Methods

The 12-month observational study was conducted on 156 patients suffering from facial ptosis, who benefited from aesthetic procedures divided into 3 groups according to the type of intervention (with minim-invasive facelifting with suspension threads, 93 people (NC); with lifting surgically, 49 persons (C); and combined, 14 persons (NC + C)). The patients presented themselves at the medical aesthetics office, where they were clinically and paraclinically evaluated. The quality of life was assessed by completing questionnaires at the beginning and end of the study period by the participants [37,38,39]. Those under psychiatric treatment, with coagulation disorders, serious heart diseases, tumor conditions, pregnant women, lauses, inflammatory and/or infectious diseases in the face were excluded from the study. Among the relative contraindications, autoimmune diseases and unbalanced diabetes were considered.

Surgical lifting is performed by folding the subcutaneous musculo-aponeurotic system (SMAS) with non-absorbable threads, anchoring the lower portion of Bichat’s Bula and excision of the skin excess.

For the minim-invasive facelift, Happy Lift-Anchorage Threads (Milano, Italy) were used; the thread thickness is 2-0 resorbable, with a size of 1 × 31.6 cm.

Combining surgical lifting with minim-invasive lifting with the minim-invasive facelift with Happy Lift-Anchorage Threads was also used; the thread thickness is 2-0 resorbable, with a size of 1 × 31.6 cm, and it is an innovative technique that offers a greater degree of stability over time of the post-interventional results [40].

### 2.1. Clinical and Paraclinical Investigation

The clinical evaluation was carried out in the medical office and the evaluation of paraclinical parameters was carried out in the authorized laboratories. At the paraclinical examinations, the coagulogram, blood pressure and blood sugar levels were checked, and the local clinical examination, rigorous anamnesis and smoking, alcohol or narcotic drugs use were taken into account. 

### 2.2. Ultrasound Examination

Ultrasound examinations using shear wave elastography were implemented for determining the elasticity of the dermis and subcutaneous cellular tissue. They were made by a radiologist with over 20 years of experience in ultrasound and over 5 years in shear wave elastography, with the help of an Aixplorer SuperSonic (Aix-en-Provence, Franța) Imagine ultrasound, Aix en Provence, France, version 11.2.0, using a linear probe with variable frequency 5–14 MHz (SL15-4). In each patient, shear wave measurements of the elasticity of the dermis and subcutaneous cellular tissue were performed at 6 locations on the face, 3 on the right half of the face and 3 on the contralateral half.

Compression with the transducer was avoided so that an appreciable gel layer remained between the transducer and the skin throughout the examination. The apparatus was set so that the values of elasticity of the skin and subcutaneous cellular tissue were displayed in kilopascals.

The place of the measurements was established by the surgical team as at the level of the zygomatic arch and at the level of Bichat’s bubble, lateral to the labial commissure. At the level of each location, 3 measurements were performed at the level of the dermis and 3 measurements immediately under the skin at the level of the subcutaneous cellular tissue. The device was set to also display the ratio between the values obtained at the level of the two samples [41]. An initial and final ultrasound examination was performed (Figure 1A initial and Figure 1B final).

### 2.3. The Rapid Vitamin D Test

Vitamin D testing was performed using the JusChek (Bucharest, Romania) rapid test, where the baseline vitamin D level is colorimetrically determined. In the cases in which the vitamin D level did not reach 30 µg, a treatment was recommended until a sufficient or optimal level. There are thus 4 categories of interpretation:-Insufficient ≤ 10 µg/mL.-Enough 30 µg/mL.-Optimal 100 µg/mL.->100 µg/mL.

Vitamin D testing was performed prior to the decision to perform the facelift. If the value of the test is insufficient, treatment with vitamin D is recommended, and after the end of the treatment, the vitamin D test is performed again. In the situation where the value of vitamin D is sufficient, it was decided to carry out the lifting intervention. 

A quick test was performed to test the hypothesis. Supplementation with vitamin D 2000 IU was recommended for 3 months; no nutritional intervention was made during this phase.

### 2.4. Statistical Analysis

Analysis was performed using the Statistical Product and Service Solutions (SPSS Inc., Chicago, IL, USA, version 20) computer software program. Demographic variables, frequency of procedures and cost data were assessed for the 2 time periods surveyed and based on 3 study lots to assess significant trends. All parameter means values, frequency ranges and standard deviations, as well as tests of statistical significance were calculated using the Student’s *t*-test and Chi square test. The Bravais–Pearson correlation coefficient was used to calculate an independent indicator of the measurement units of the two variables. A value of *p* < 0.05 was assigned to statistical significance, as was *p* < 0.01 to high-level statistical significance. Post hoc analysis (Bonferroni) was used to analyze differences between groups, as an additional subgroup analysis.

## 3. Results

### 3.1. Demographic Description

The cohort of 156 people with an average age of 50.92 ± 10.60, between min 35 and max 75 years old, predominantly women, from an urban environment with higher education, shown in Table 1, presented themselves in the medical office. The patients in the study were divided into three groups according to the type of intervention performed:

Group 1—NC—Those who underwent minim-invasive procedures with anchoring wires (93 people).

Group 2—C—Those with surgical intervention (49 people).

Group 3—NC + C—Those with combined minim-invasive and surgical intervention (14 people), shown in Figure 2.

**Table 1 healthcare-11-01490-t001:** Demographic description of the cohort. Descriptive statistics.

Parameters		Groups						Total	
		NC		C		NC + C			
		*n*	%	*n*	%	*n*	%	*n*	%
Gender	Men	27	17.3	6	3.8	0	0.0	33	21.15
	Women	66	42.3	43	27.6	14	9.0	123	78.85
Environment of provenance	Urban	80	51.3	45	28.8	14	9.0	139	89.10
	Rural	13	8.3	4	2.6	0	0.0	17	10.90
Education	Secondary education	41	26.3	6	3.8	0	0.0	47	30.12
	High education	52	33.3	43	27.6	14	9.0	109	69.88
Age		45.76 ± 8.09		60.71 ± 8.41		66.50 ± 5.79		50.92 ± 10.60	
Total		156							

*n* = number of patients, NC = minim-invasive group, C = surgical group, NC + C = surgical lifting combined group.

### 3.2. Facelifting

These days, when people’s life expectancy is increasing, and youth and beauty are actively promoted everywhere, aesthetic medicine is in high demand. One of the main tasks becomes the elimination of the external signs of aging of the body; therefore, the last years have been characterized by an increase in interest in the methods used for improving the facial appearance with the repositioning of ptosis tissues. Figure 3 shows the appearance of the zygomatic region before and one year after the facelifting intervention. Before: moderate facial ptosis in the mandibular and zygomatic region with emphasis on the nasolabial folds. In Figure 3, one can see the results obtained in the zygomatic and mandibular region by the minimally invasive technique.

### 3.3. Minim-Invasive Facelifting with PLA-CL Suspension Wires for Medical Purposes

Subcutaneous cellular tissue and dermis changes following a facelift in a patient with left hemiparesis (Figure 4).

In one patient in the anterior image with left hemiparesis, the thickness of the subcutaneous cellular tissue and dermis was measured at the level of both hemifaces. Table 2 shows changes in the subcutaneous cellular tissue and the dermis following the minim-invasive lifting with anchoring wires of the left hemiface.

Figure 5 shows the changes in the subcutaneous cellular tissue and the dermis in both the right and left hemispheres. A significant difference between the two parts can be observed regarding the structure and subcutaneous thickness.

### 3.4. Combined Lifting

Surgical lifting is performed by folding the subcutaneous musculo-aponeurotic system (SMAS) with non-absorbable threads, anchoring the lower part of the fatty body of the cheek (Bichat’s bubble), excision of the skin excess and insertion of PLA-CL suspension threads (Figure 6).

In Table 3, you can follow the measurements of the shear wave elastography in the combined intervention.

Figure 7 shows a graphical representation of the measured differences in the mandibular line on the left and right, initially and finally.

The facial aspect regarding the mandibular line is shown in Figure 8, both from the right side before and after (Figure 8A,B) and from the left side (Figure 8C,D) through the combined surgical technique.

The elasticity of the skin was measured in each of the patients, and the comparisons of the results were based on these measurements. The clinical results were verified by elastography, after which the study was performed and the results were interpreted. A coefficient was calculated from the data obtained according to Young’s modulus (E) [42], E = (k = (Ʃ x + Ʃy) initial − (Ʃx + Ʃy) final) [43]. Significant differences were observed in terms of the three research groups, between NC and C (*p* = 0.012), between NC and NC + C (*p* = 0.001) and between C and NC + C (*p* = 0.001).

From Figure 9, it can be seen that more people with risk factors can be included in the minim-invasive procedure because the least invasive technique is used, which involves fewer operative risks and anesthetics and the absence of skin incisions on the face. In the surgical procedure and in the combined ones, there are only six patients with hypertension, six who consume alcohol, only two smokers and three patients with diabetes. The real advantage is for diabetics and hypertensives, as the minimally invasive procedure is recommended for the first time.

### 3.5. Risk Factors

Additionally, a history of postoperative wound infection or known colonization with methicillin-resistant Staphylococcus aureus must be excluded. High-risk patients should be screened for MRSA colonization and a decolonization protocol used preoperatively, as this has been shown to significantly reduce surgical site infection (SSI) [44].

Among the modifiable factors, we followed smoking, alcohol consumption, narcotic substances, hypertension or unbalanced diabetes, as can be seen from Table 4.

### 3.6. Vitamin D3

Typically, the dense extracellular structure in fibrotic tissues is described as the extracellular matrix (ECM) or simply as collagen. There is good and bad collagen in fibrosis and only a change in location can change the function from good to bad [15]. While type IV basement membrane collagen anchors epithelial cells and other cells in a polarized manner, types I and III interstitial fibroblast collagen do not provide directional information. Furthermore, feedback loops from biologically active degradation products of some collagens exemplify the importance of having the right collagen in the right place at the right time when it comes to controlling cell function, proliferation, matrix production and fate. Examples are type VI interstitial collagen and type XVIII basement membrane collagen. There are 28 known types of collagen, and it is important to keep track of the collagen type especially in minimally invasive interventions. It is assumed that the collagen formed as a result of fibrosis in the extracellular matrix, in the presence of vitamin D, changes its location in the basal membrane and will have a supporting role. Following this hypothesis, the level of vitamin D in patients was followed before, during and after the intervention. In Table 5 and Figure 10, the baseline and final level of vitamin D are presented.

### 3.7. Correlations

Following the evolution of the thickness of the dermis and the subcutaneous cellular tissue through the initial and final elastography measurements, a coefficient was calculated for each table (k = (Ʃx + Ʃy) initial − (Ʃx + Ʃy) final) in order to compare the development. A decrease in elasticity due to fibrosis can be observed in the NC group; in C, the increase in elasticity is according to the speed of aging; and in the combined technique, the elasticity is lower, with minor variations. A non-significant difference (*p* > 0.05) can be observed between those with sufficient and optimal vitamin D in the NC group, as in the NC + C group. There was a substantial difference found in terms of skin elasticity in the C group between those with sufficient vitamin D and those with the optimal level. In those who had enough, there was a greater increase in elasticity; in those who had optimal vitamin D, the average difference was smaller. The difference regarding elastography is presented in Figure 11.

## 4. Discussion

### 4.1. Clarification of the Nature and the History of Concerns Related to the Occurrence of Facial Ptosis

Patients describe their concerns related to the facial image, the degree of dissatisfaction and their expectations related to improving their facial appearance. The patient’s perception of “abnormality” in appearance is subjective and aesthetic doctors should make a visible association between the severity, the defect perceived by the patient and their degree of emotional distress [45]. Before making the decision to perform the facelift, the clinician should investigate the impact of the intervention on the patient’s psychological well-being and daily routine (including on their social, intimate and professional lifestyle). The amount of time spent thinking about a certain feature, as well as the frequency of facial appearance checks (using a mirror or other reflective surfaces) over a typical day, should be investigated. If the degree of distress appears grossly disproportionate to the perceived defect, if mirror checking appears excessive and/or if their perceived concerns with appearance interfere with daily activities or affect a relationship, the patient should be considered for an evaluation/a more detailed assessment [39,46]. In our study, the patient with left hemiparesis was subjected to a minimally invasive intervention. After the intervention, from the graphic comparison of the thickness differences between the left and right hemiface initially and finally, a balancing of the subcutaneous tissue can be observed. This can be explained by the reduction in fat mass due to fibrosis formation. 

In order to stabilize the results, it is important to maintain an optimal level of vitamin D. As adjuvant therapies, which can improve the results, are the correct maintenance of the skin, the use of innovative cosmetic products (especially formulated considering the nanocarriers [47] and also those plant-based, rich in bioactive compounds [48,49,50] and easily penetrating the skin) and regular visits to the dermatologist and/or esthetician should occur.

### 4.2. Patient Motivation for Requesting the Procedure

Studies have found that patients who are motivated to seek cosmetic surgery for intrinsic reasons (e.g., to improve self-confidence) are more likely to be satisfied with their postoperative results than those motivated by extrinsic factors (e.g., to win a new romantic partner or get a job promotion) [45].

When it comes to evaluating the motivation to seek surgery, Ref. [39] it is also recommended to address the following topics:When did the prospective patient start thinking about surgery to alter their appearance? Was there a particular triggering event? Are there specific influences that may influence their outcome expectations in unrealistic ways?Has the prospective patient done anything else to improve their appearance? These questions should elicit information about any unusual or maladaptive behaviors that exist in relation to their physical appearance [51] to be documented examples of individuals with BDD attempting to modify their appearance on their own.Why is the prospective patient interested in surgery at this time? What factors led the patient to request a consultation at this time? Is it in response to a significant life event (e.g., divorce) or a series of events? If so, the clinician can explore the feasibility of the proposed surgical intervention in response. Is the future patient’s motivation for surgery a result of intrinsic or extrinsic factors?

### 4.3. Understanding of the Procedure by the Future Patient

The clinician should assess the prospective patient’s understanding of the potential risks (both physical and psychological) and complications involved in the procedure(s) in question. Because the patient’s processing of the information provided is likely to be influenced by cognitive biases (for example, media-derived biases about the process and likely outcomes of cosmetic surgery), it is useful to assess their understanding by asking them to summarize their expectations about how the operation will proceed and the risks involved.

Most facial features showed a dose-dependent relationship between pack smoking history and aging severity. Higher alcohol consumption also increased the severity of some signs of facial aging. Heavy alcohol consumption (eight or more drinks per week) was associated with increased severity of almost all facial features [52]. In our study, the most smokers were registered in the C group and the fewest in the one with the combined technique. Among moderate drinkers, only midface volume loss and under-eye puffiness were associated with alcohol consumption. Furthermore, only visible blood vessels were significantly increased in heavy drinkers compared to moderate drinkers and in wine-only drinkers compared to non-drinkers, supporting previous findings that alcohol use is linked to rosacea [53]. Alcohol abuse has been reported to reduce fat mass, which may underlie the loss of midface volume reported by heavy drinkers. The increase in puffiness under the eyes and dark circles may have been due to the exposure of the suborbital fat layer as the midface volume receded [54]. Although the return recommendation is long for minimally invasive operations and just for retouching after the other two interventions, an accelerated aging process associated to risk factors (smoking or alcohol intake) can considerably shorten this interval. The least drinkers were in the combined technique group in our study.

Facelift techniques can be classified according to the depth and extent of the dissection applied. Subcutaneous musculoaponeurotic system (SMAS) nesting and tucking called “SMAS lifts” or “classic facelifts” are the most commonly used techniques. Folding techniques involve folding the SMAS inward and suture suspension without any SMAS incision, while nesting techniques involve an SMAS incision with a portion of the SMAS either removed or transposed with or without limited sub-SMAS dissection. Aging changes in the lower part of the face and neck can be treated successfully with SMAS lifting techniques. In any case, there is no lifting effect in the middle of the face or improvement in nasolabial folds because the retaining ligaments (zygomatic skin and masseteric skin) that prevent the transmission of traction to the malar portion of the facelift dissection are not released. Thus, the worst results were recorded with the classic technique, a fact that led to the need for additional techniques to make the facelift more effective.

Extended facelift techniques involve combined minimally invasive surgery and produce a combined, balanced and harmonious rejuvenation of the midface, cheek and lower face without requiring a separate midfacelift procedure. There are different techniques that have similar extended dissections in the midface, with some variations: The extended SMAS technique involves a long skin flap and a distinct SMAS flap dissected and pulled separately. The high SMAS technique has a similar dissection but involves a higher SMAS flap along the upper border of the zygomatic arch. The deep-plane facelift involves undermining the SMAS skin flap as a single unit after a more limited subcutaneous dissection. In the facelift composite plane, in addition to the deep-plane facelift dissection, the inferior part of the orbicularis oculi muscle is also dissected and included in the flap. Because they have a single unit, deep and composite flap lifts allow for excellent blood supply to the overlying skin [55]. A modification of the SMAS lift and suspension is described with the strategic submalar fold, providing a powerful technique for primary facial rhytidectomy that provides reliable and significant malar lift, orbicular suspension, gingival relief and repositioning of the oral commissure [56,57,58,59].

Vitamin D has an important role in the mineralization of the collagen matrix [60]. Vitamin D3 is created endogenously in the skin, and a lack of it is associated not only with osteo-musculoskeletal disorders but also with a number of chronic conditions and diseases [32]. The protective effect of vitamin D at the joint level, intervening in collagen synthesis, was also recorded [61]. Low levels of vitamin D are also associated with the loss of cartilage in the joints [62], collagen formation being blocked. In the last decade, studies on the role of vitamin D have increased, including skin senescence [1,63,64]. Looking at the recommended dose in deficiencies, or prevention, has many variations and is still being studied. That is why the maintenance dose is recommended.

### 4.4. Limitations and Strengths of the Study

One of this work’s shortcomings is the relatively limited time spent observing the patients. The real results can be reported after those 6 years, which the description of the minimally invasive lifting theoretically provides. Another limitation may be vitamin deficiencies (niacinamide, vitamin B6, biotin and pantothenic acid), where the deficiencies may interfere with the loss of skin elasticity and the collagen deposited, and the lack of laboratory data on the exact level of vitamin D, using the rapid test. The season when the patient presented was not taken into account either, which can still modify the results. These aspects (nutrition, vitamin K2 and season) should be followed in future studies. Another limitation of the study may also represent internal and external factors that change the elasticity of the skin. 

The strengths of the study are the verification and correlation of the loss of skin elasticity with the serum level of vitamin D. This could be a starting point in revolutionizing facelifting, by making the interventions more efficient.

## 5. Conclusions

Insignificant differences were registered between those with a sufficient vitamin D level and those with an optimal level in the NC group. There was a significant decrease in the subepidermal layer. This difference can be explained by the fibrosis formed in the first phase. The presence of vitamin D, at least enough, intervenes in changing the location of collagen, thus ensuring the skin’s elasticity over time. In patients with classic surgical intervention, there was an increase in the subcutaneous volume that leads to a decrease in the elasticity of the subcutaneous tissue.

The fibrosis formed was lower in this group, which can be explained by the acute inflammation present due to the surgical intervention, which causes the reduction in profibrotic expression. The best results were obtained with the combined technique; where the difference in the volume of the skin is small, the subcutaneous volume did not increase. A minimal intake of vitamin D and a minimal vitamin D level are necessary for achieving long-lasting results.

## Figures and Tables

**Figure 1 healthcare-11-01490-f001:**
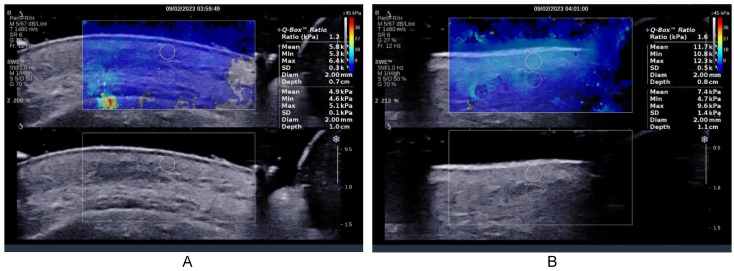
Elastography of the soft parts before the intervention minim-invasive. The sample used had a size of 2 mm. This was initially placed at the level of the dermis, and then immediately below the dermis, adjacent to the original sample. (**A**) After one week, the final result for each location was the arithmetic mean of the 3 measurements at the level of the dermis, respectively, of the 3 subdermal measurements, respectively, the arithmetic mean of the ratios obtained at the 3 measurements (**B**).

**Figure 2 healthcare-11-01490-f002:**
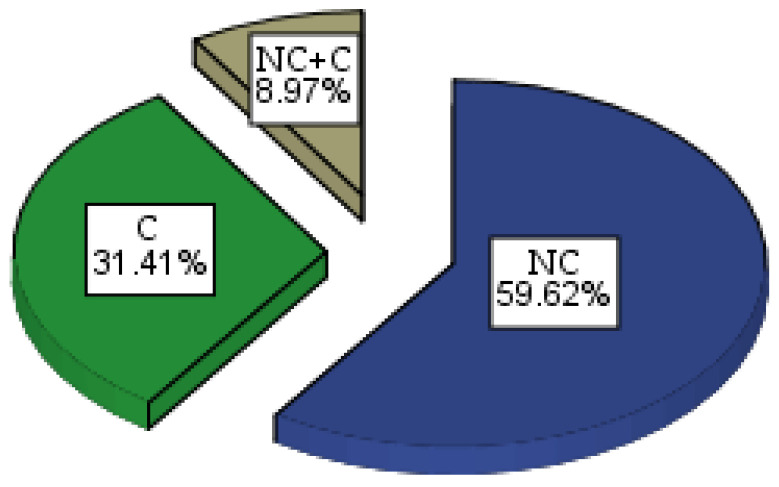
Graphical representation by the pie method of percentage distribution of patients in the 3 study groups.

**Figure 3 healthcare-11-01490-f003:**
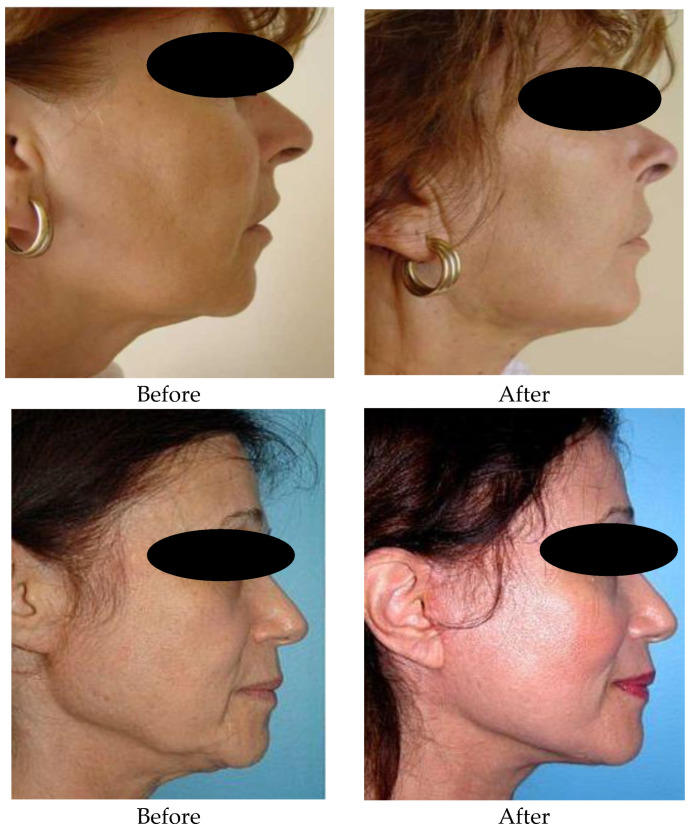
Reconstruction of the zygomatic area before and after 1 year, through the technique of minim-invasive facelifting with PLA-CL suspension wires. In both cases 3 wires were used. The 1st thread was mounted at the level of the zygomatic region, the 2nd at the level of the genial region which also targeted the labial commissure, the 3rd thread was mounted at the level of the mandibular region.

**Figure 4 healthcare-11-01490-f004:**
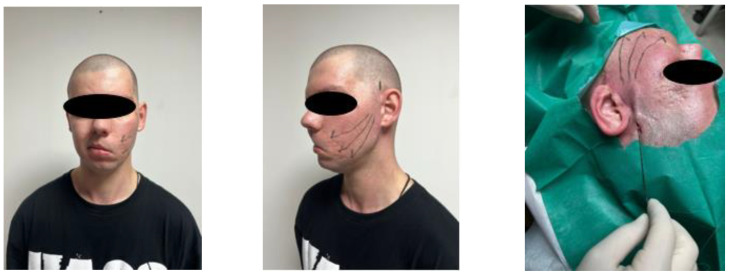
Facelifting in a patient with hemiparesis. In the image, you can see the regions drawn where the wires were mounted. Happy Lift-Anchorage Threads were also used; the thread thickness is 2-0 resorbable, with a size of 1 × 31.6 cm.

**Figure 5 healthcare-11-01490-f005:**
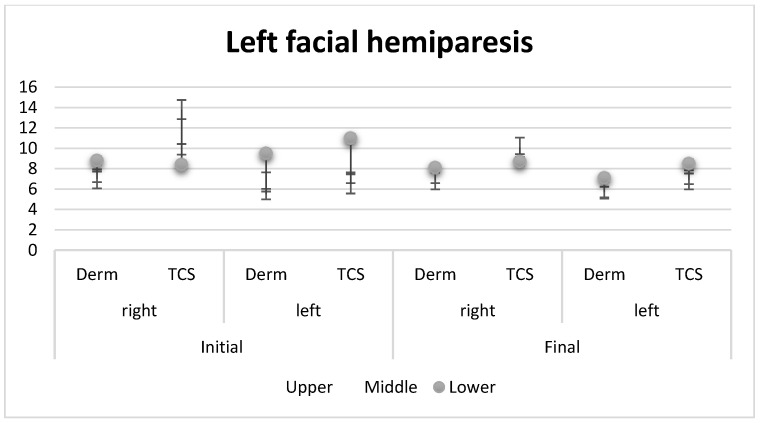
Graphic comparison of the thickness differences between the left and right hemiface, by stock method, in changes in subcutaneous cellular tissue and dermis following nonsurgical left hemifacelift. Derm = dermis cellular tissue, TCS = subcutaneous cellular tissue.

**Figure 6 healthcare-11-01490-f006:**
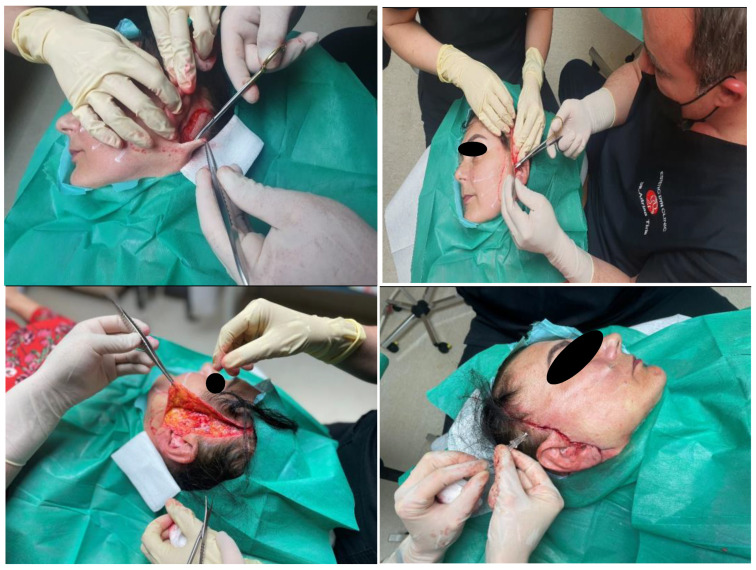
Surgical lifting combined with PLA-CL suspension threads. A surgical facelift with SMAS anchoring and a non-surgical facelift with Happy Lift-Double Niddle anchoring wires were performed in the same session; the thickness of the 2-0 resorbable wire, thread model × 12 cm.

**Figure 7 healthcare-11-01490-f007:**
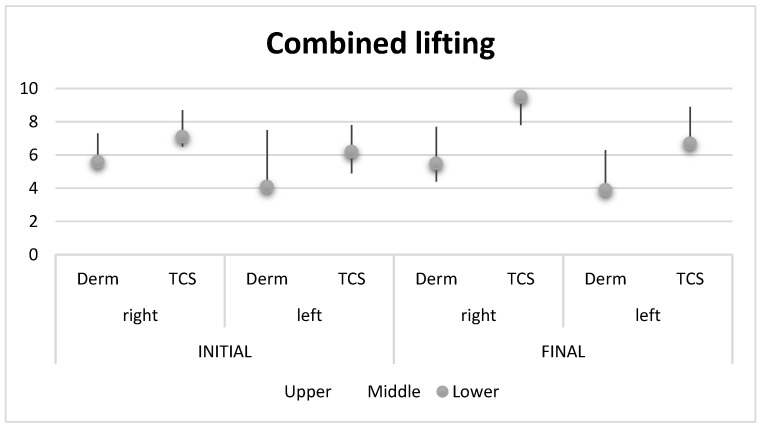
Graphic presentation of initial and final left and right mandibular line differences, by stock method, in shear wave elastography measurements, in the combined intervention.

**Figure 8 healthcare-11-01490-f008:**
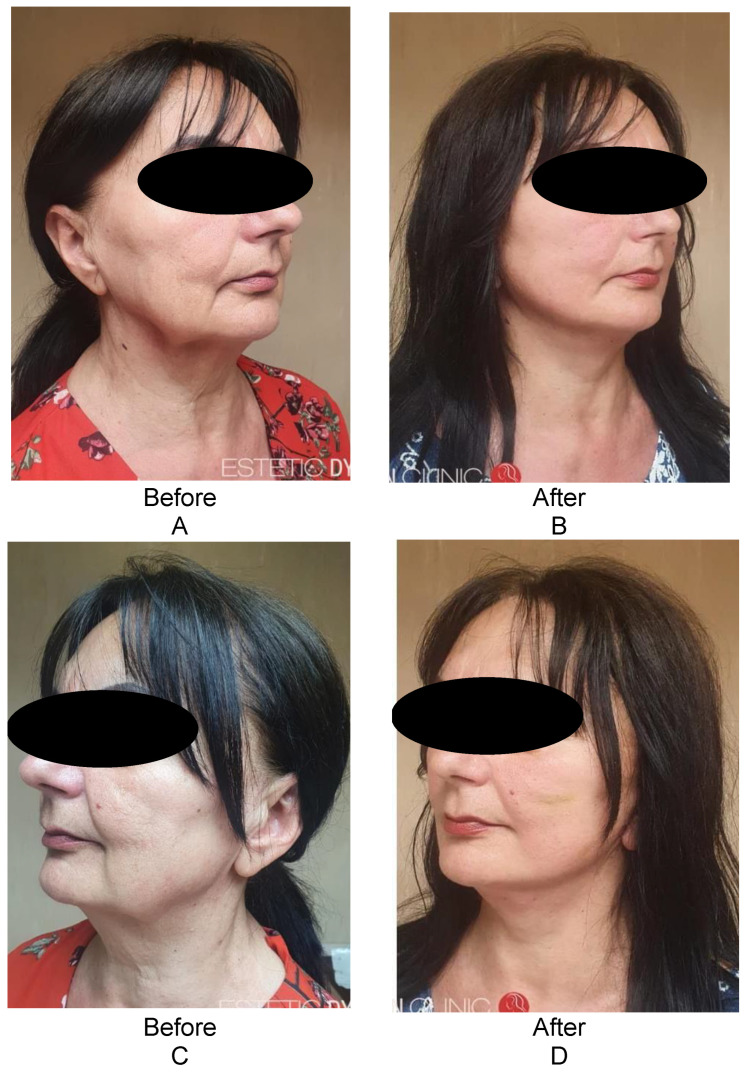
The facial appearance regarding the mandibular line from the right side before and after (**A**,**B**) and from the left side (**C**,**D**) by the hybrid technique. A surgical facelift with SMAS anchoring and a non-surgical facelift with Happy Lift-Double Niddle anchoring wires were performed in the same session; the thickness of the 2-0 resorbable wire, thread model × 12 cm.

**Figure 9 healthcare-11-01490-f009:**
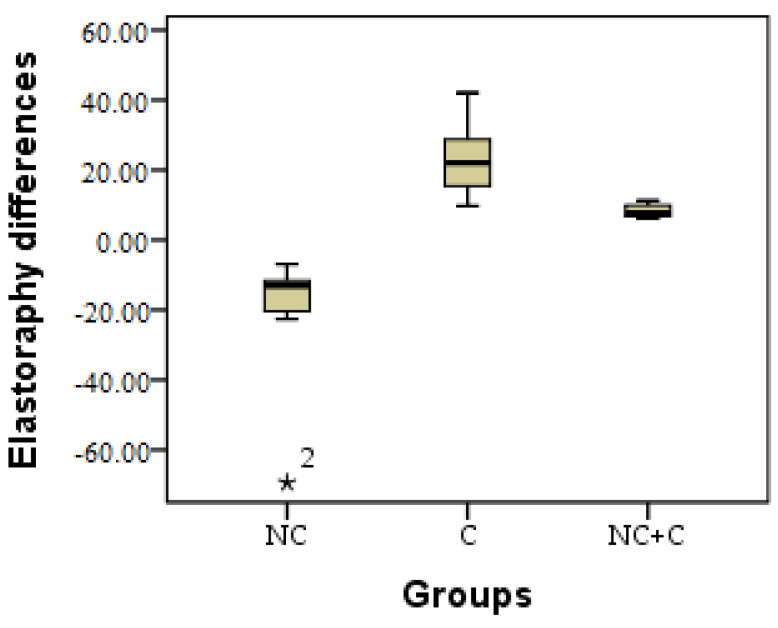
Graphical representation by the boxplot method of results of the initial–final elastography in the 3 research groups. * = excepting.

**Figure 10 healthcare-11-01490-f010:**
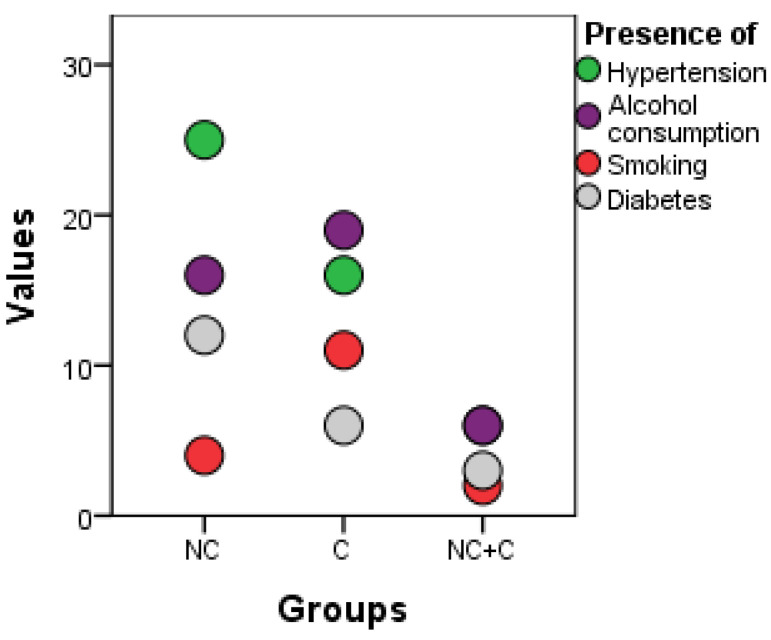
Graphic representation of the risk factors in the 3 research groups by dots methods. NC = minim-invasive group, C = surgical group, NC + C = surgical lifting combined group.

**Figure 11 healthcare-11-01490-f011:**
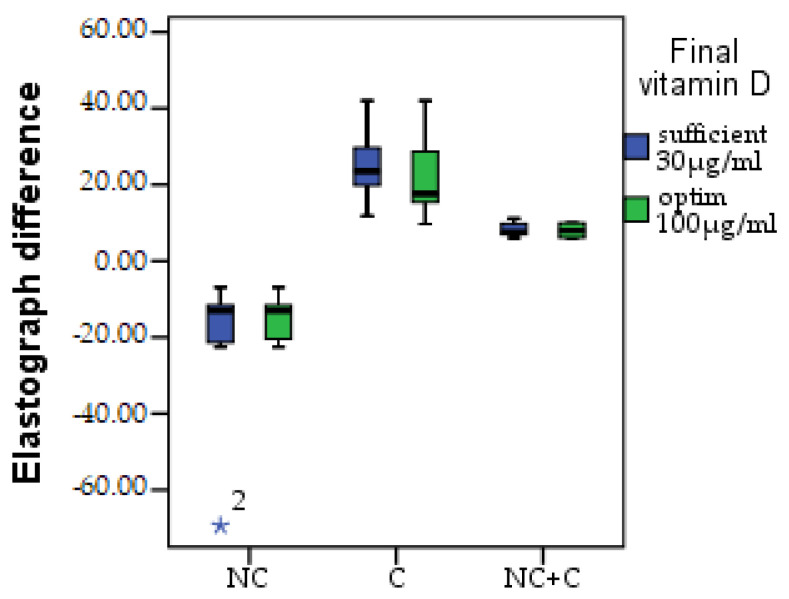
Elastography difference, NC = minim-invasive group, C = surgical group, NC + C = surgical lifting combined group. * = excepting.

**Table 2 healthcare-11-01490-t002:** Changes in subcutaneous cellular tissue and dermis following nonsurgical left hemifacelift.

Parameters		Upper	Middle	Lower
Before				
Right	Derm	7.2 ± 0.6	7.0 ± 0.4	8.8 ± 1.1
	TCS	9.9 ± 0.8	13.8 ± 3.5	8.4 ± 0.8
Left	Derm	5.5 ± 0.4	6.7 ± 0.8	9.5 ± 0.5
	TCS	7.1 ± 0.2	6.5 ± 0.8	11.0 ± 1.3
After				
Right	Derm	7.1 ± 0.5	6.9 ± 0.7	8.1 ± 1.4
	TCS	8.9 ± 0.6	10.1 ± 2.5	8.7 ± 0.6
Left	Derm	5.7 ± 0.4	6.0 ± 0.6	7.1 ± 0.8
	TCS	7.0 ± 0.3	6.9 ± 0.5	8.5 ± 1.1

Derm = dermis cellular tissue, TCS = subcutaneous cellular tissue.

**Table 3 healthcare-11-01490-t003:** Shear wave elastography measurements in the combined intervention.

Before				
Parameters		Upper	Middle	Lower
Right	Derm	5.5 ± 0.7	7.3 ± 1.2	5.6 ± 0.7
	TCS	6.5 ± 0.4	8.7 ± 1.1	7.1 ± 0.5
Left	Derm	6.1 ± 0.4	7.5 ± 0.8	4.1 ± 1.0
	TCS	4.9 ± 1.1	7.8 ± 1.2	6.2 ± 0.6
After				
Right	Derm	7.7 ± 1.0	4.4 ± 0.4	5.5 ± 0.6
	TCS	9.5 ± 0.6	7.8 ± 1.1	9.5 ± 0.5
Left	Derm	6.0 ± 1.0	6.3 ± 0.9	3.9 ± 1.0
	TCS	8.9 ± 0.7	7.2 ± 0.7	6.7 ± 0.7

Derm = dermis cellular tissue, TCS = subcutaneous cellular tissue.

**Table 4 healthcare-11-01490-t004:** Risk factors in the 3 research groups at the start of the study period.

Parameters		Groups					
		NC		C		NC + C	
		*n*	%	*n*	%	*n*	%
Hypertension	No	68	43.6	33	21.2	8	5.1
	Yes	25	16.0	16	10.3	6	3.8
Alcohol consumption	No	77	49.4	30	19.2	8	5.1
	Yes	16	10.3	19	12.2	6	3.8
Smoking	No	89	57.1	38	24.4	12	7.7
	Yes	4	2.6	11	7.1	2	1.3
Diabetes melitus	No	81	51.9	43	27.6	11	7.1
	Yes	12	7.7	6	3.8	3	1.9

NC = minim-invasive group, C = surgical group, NC + C = surgical lifting combined group, *n* = number of patients.

**Table 5 healthcare-11-01490-t005:** Descriptive presentation of baseline and final vitamin D levels.

Vitamin D		Groups					
		NC		C		NC + C	
		*n*	%	*n*	%	*n*	%
Initial	Insufficient < 10 µg/mL	0	0.0	0	0.0	0	0.0
	Sufficient 30 µg/mL	17	18.3	0	0.0	14	100.0
	Optim 100 µg/mL	76	81.7	49	100	0	0.0
	Exces > 100 µg/mL	0	0.0	0	0.0	0	0.0
Final	Insufficient < 10 µg/mL	0	0.0	0	0.0	0	0.0
	Sufficient 30 µg/mL	15	16.1	7	14.3	12	85.7
	Optim 100 µg/mL	78	83.9	42	85.7	2	14.3
	Exces > 100 µg/mL	0	0.0	0	0.0	0	0.0

NC = minim-invasive group, C = surgical group, NC + C = surgical lifting combined group, *n* = number of patients.

## Data Availability

All the data processed in this article are part of the research for a doctoral thesis, being archived in the aesthetic medical office, where the interventions were performed.

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
