# Peer review of "Can Vitamin D Levels Alter the Effectiveness of Short-Term Facelift Interventions?"

_healthcare, 2023, doi:10.3390/healthcare11101490_

Round 1

Reviewer 1 Report

Comments and suggestions are set out in the Annex

Author Response

Firstly, we, the authors of the present manuscript wish to thank you for thoughtful commentary you have provided to improve the quality of the paper. I am very grateful for the time and effort you have devoted to this task. We have extensively revised my manuscript according to the recommendations. All changes in the text and the new figures that we have redesigned are highlighted. Please, see the point-by-point answers to your comments below. All correction was highlighted in the manuscript.

Reviewer 2 Report

The research needs  a clear biochemical  background,  adeguate methods and statistical power. 

Author Response

(The authors gave the same response as above.)

Reviewer 3 Report

You present findings of clinical and paraclinical interventions to improve face-lift outcomes in ageing patients. This is an important extension to the classic surgical intervention. Elsewhere you present literature, findings and interpretations, as well as discussed and concluded in as scientific as this level of study could go. However, I found your abstract to be lacking coherence, as the statements therein are too brief. As well, you need to expand your figure legends to include key experimental information including a brief description of the procedure, description of elements in the figure etc.

I found the work exciting and scientifically sound. 

Author Response

(The authors gave the same response as above.)

Reviewer 4 Report

This work describes a probable association between vitamin D3 levels and face lifting (with surgical and non surgical techniques), to analyze a possible interaction between subcutaneous volume and vitamin D3 concentration. Authors have showed that patients with normal levels of this hormone and lifting results.

I think that this work is novelty and well written, and there are no significant changes required.

Author Response

First of all, we, the authors of this manuscript, would like to thank you for your comment. I am very grateful for the time and effort you put into this task. We are marked by your appreciation and support. We hope that the improvements we made to the paper will only increase the value of this research. Once again, thank you for the favorable comment!

Reviewer 5 Report

Dear authors,

Some comments, though:
- the whole article needs to be extensively edited for the English language and the sentences are too long. Use the KISS-principle: "Keep It Short and Simple".

- Abstract: Aim for the reader is not clear. Statistic values are missing. Link between Vit. D and Face-Lifts is not clear.

- The introduction is insufficient in regard to Vitamin D (e.g. Vit. D in surgery). No literature is mentioned which supports the title or aim of this study. Hereof, scientific aspect / base is lacking in this study. A common thread is missing. 

- Materials and Methods are insufficiently reported. Line 87 the authors mentioned that quality of life was assed. There is no information what for a test was used / no reference / no data in the results section. Line 90: what is "lauses"? Line 96: "The non-surgical (minimally-invasive.." A contradiction. Line 87: what is paraclinical in this case? Line 96: Abbrevation PLA-V? 

- Statistical analysis is redudant. Is there a difference between Student's t-Test and  independent / paired t-Test? Same wording please.

- Elastography is reported insufficiently (e.g. software is missing, unprecise description of Figure 1,misleading information wihtin figure 1: "breast / superficial breast", missing description what the reader can see in figure 1 exactly -> anatomical structures, exakt anatomical measuring points).  Who did the authors achieve a standardized evaluation with elastography?

- Patients demographics are insufficient (only gender, education, environment of provenance, age) to deduce this kind of research question (see main title of this manuscript).

- Tables and figures are a mess. Abbrevations are often not explained. Descriptions are partly missing - for the reader the figures and tables are not comprehensible. All figures were created from SPSS, this is a scientific journal / manuscript and no essay in school. There are standards in scientific reporting - these standards are not fulfilled here.  

- Results: no p- values are reported. It s not clear which test was used, or which parameter superior / inferior. The "3.6 Vitamin D" section is more an introduction than a results part. The Vitamin D intake etc. is not reported at all - it seems that the authors didn't control anything in regard to Vitamin D. Bevor surgery and 12 months after surgery Vitamin D level was evaluated. Based on this, the authors conclude impact/effects of Vitamin-Level on face-lift outcomes - again, without any literature regarding this and without an appropriate study design. 

- Discussion: the discussion mostly describes or "discusses" the face-lift-associated points, again, without linking Vit. D to face-lifts in a scientific manner. The discussed part in regard to Vitamin D is very scarce, although the title implies something different. 

Taken together, this manuscript has a lot of serious flaws in every section. The study design and the results are doubtful, thus, the conclusion in regard to the Vitamin D effect on the outcome after 12 months post-surgery is problematic. Hence, i can't recommend this manuscript for publication.

Author Response

First of all, we, the authors of this manuscript, would like to thank you for the thoughtful comment you provided to improve the quality of the paper. We are very grateful for the time and effort you put into this task. I understand that in its previous format it does not meet the criteria for publication. We revised our manuscript extensively as recommended. All the changes in the text and the new figures that we have redesigned are highlighted. Please see point by point responses to your comments below. All corrections have been highlighted in the manuscript.

Round 2

Reviewer 1 Report

Thank you that the manuscript has been corrected according to my suggestions and comments. The authors supplemented the information, which increases the substantive value of the work. I accept the publication in this form.